# Microstructural Changes and Determination of a Continuous Cooling Transformation (CCT) Diagram Using Dilatometric Analysis of M398 High-Alloy Tool Steel Produced by Microclean Powder Metallurgy

**DOI:** 10.3390/ma16124473

**Published:** 2023-06-19

**Authors:** Michal Krbata, Robert Ciger, Marcel Kohutiar, Maros Eckert, Igor Barenyi, Bohdan Trembach, Andrej Dubec, Jana Escherova, Matúš Gavalec, Naďa Beronská

**Affiliations:** 1Faculty of Special Technology, Alexander Dubcek University of Trenčín, 911 06 Trenčín, Slovakia; michal.krbata@tnuni.sk (M.K.); robert.ciger@tnuni.sk (R.C.); maros.eckert@tnuni.sk (M.E.); igor.barenyi@tnuni.sk (I.B.); jana.escherova@tnuni.sk (J.E.); matus.gavalec@tnuni.sk (M.G.); 2Private Joint Stock Company, Novokramatorsky Mashinostroitelny Zavod, 84305 Kramatorsk, Ukraine; btrembach89@gmail.com; 3Faculty of Industrial Technologies, Alexander Dubcek University of Trenčín, 020 01 Púchov, Slovakia; andrej.dubec@tnuni.sk; 4Institute of Materials and Machine Mechanics, SAS, Dúbravská cesta 9/6319, 845 13 Bratislava, Slovakia; nada.beronska@savba.sk

**Keywords:** powder metallurgy, dilatometry, heat treatment, carbide

## Abstract

The paper deals with the dilatometric study of high-alloy martensitic tool steel with the designation M398 (BÖHLER), which is produced by the powder metallurgy process. These materials are used to produce screws for injection molding machines in the plastic industry. Increasing the life cycle of these screws leads to significant economic savings. This contribution focuses on creating the CCT diagram of the investigated powder steel in the range of cooling rates from 100 to 0.01 °C/s. JMatPro^®^ API v7.0 simulation software was used to compare the experimentally measured CCT diagram. The measured dilatation curves were confronted with a microstructural analysis, which was evaluated using a scanning electron microscope (SEM). The M398 material contains a large number of carbide particles that occur in the form of M_7_C_3_ and MC and are based on Cr and V. EDS analysis was used to evaluate the distribution of selected chemical elements. A comparison of the surface hardness of all samples in relation to the given cooling rates was also carried out. Subsequently, the nanoindentation properties of the formed individual phases as well as the carbides, where the nanohardness and reduced modulus of elasticity (carbides and matrix) were evaluated.

## 1. Introduction

The dilatometric and plastometric analysis using modern dilatometers is currently a widespread method for evaluating microstructural changes and selected properties (hardness and magnetic properties) of metallic materials. In the case of dilatometric analysis of steels, current research focuses mainly on the study of phase transformations such as austenitizing during heating, but also the transformation of austenite during cooling [1,2,3]. During the heat treatment of steels, phase transformations occur. During heating, austenitization occurs, and after subsequent cooling, austenite disintegrates into martensite, bainite, pearlite, ferrite, or a combination of individual phases. The most important factor influencing the formation of individual phases is the cooling rate; the secondary factor is the height of the austenitizing temperature and the chemical composition of the given material. This issue is described by Ferrari, C. et al. [4]. It is possible to determine the proportions of individual phases using electron microscopy, but microscopy does not provide information under which conditions the transformation occurred. For this reason, dilatometry is a suitable tool for investigating phase transformations [5,6]. In principle, the dilatometric device performs a precise measurement of length change in the order of µm during simulated heat treatment [7,8]. The phase change is subsequently manifested as a step change in the resulting temperature–dilatation dependence [9,10,11,12].

As mentioned above, the result of the dilatometric analysis is the dilatation curves for individual cooling rates. By evaluating the curves in the area of step changes, we obtain exact temperatures representing the beginning and the end of the individual phase's formation. These data can then be processed into the resulting CCT diagram, which is necessary for the heat treatment of the given steel in the industry [13,14,15,16].

Currently, the still progressive method of steel production is the powder metallurgy process [17,18]. From an energy point of view, this process is several times more demanding than conventional foundry technologies, in which energy consumption is, in principle, limited only to the melting of basic materials and casting into ingots [19,20,21]. However, the resulting steel created by powder metallurgy may have properties far exceeding conventionally produced steel in terms of hardness, strength, and corrosion resistance [22,23,24]. The process of steel production by powder metallurgy includes the melting of basic materials, atomization (production of powders), and subsequent sintering [25,26,27].

The examined steel that we will deal with in this article is called BÖHLER M398; it is a chrome martensitic steel produced by the powder metallurgy process. The potential use of this steel is to replace the older steel M390 in the plastics industry, in particular as a material for the production of screw conveyors [28,29]. During operation, screws are exposed to increased temperature, mechanical stress, and an aggressive chemical environment [30,31,32,33,34]. For this reason, increased requirements are placed on the material of the screws [35,36,37]. The implementation of M398 steel in the production of screws is expected to have an effect in the form of increased operational capacity as well as production in terms of the use of plastics with a higher glass content.

There is not a single work in the current literature that deals with the issue of determining the CCT diagram of the studied M398 steel or describes the influence of the cooling rate on the change in the appearance of possible phases. BÖHLER, as the exclusive importer and manufacturer of this steel, does not provide any information regarding the microstructural behavior study. For this reason, we decided to conduct this research and compare our experimental results with the JMatPro^®^ API v7.0 simulation software. Since this tool steel is planned to be used as a modern replacement for injection molding screws, with these screws, it is necessary to thoroughly analyze the resulting phase composition in order to reduce the rate of wear.

The examined steel has a high chromium content, which contributes not only to the increase in hardness and wearability but also to corrosion resistance—the high chromium content results in the formation of carbides, primarily chromium-based M_7_C_3_ carbides, and MC carbides. The BÖHLER company states a percentage of ~25% M_7_C_3_ and ~5% MC.

The occurrence of carbides in the structure of materials from a general point of view inhibits the growth of austenitic grains, which, last but not least, has a positive effect on the resulting mechanical properties [38,39,40,41,42]. Since the experimental steel contains more than 30% carbides in the total volume, we can assume that the grain growth will not depend on the cooling rate or only to a minimal extent [43,44,45,46].

The main goal of the present contribution will be to observe the microstructural changes and the construction of the CCT diagram using the dilatometric analysis of the high-alloy tool steel M398, which is produced by powder metallurgy. This CCT diagram will also be compared with the CCT diagrams from the JMatPro^®^ API v7.0 simulation software. Sub-objectives will consist of EDS map analysis of selected elements, nanoindentation evaluation of carbide particles, and formed phase matrices, where nanohardness and reduced modulus of elasticity were evaluated.

## 2. Materials and Methods

### 2.1. Experimental Material and Procedure

The experimental material M398, developed by BÖHLER, aims to meet high demands in plastics processing. It is a martensitic chrome steel with a high carbon content produced by the powder metallurgy method. Thanks to the production method and chemical composition, the steel provides extremely high resistance to mechanical wear and corrosion. The prerequisite for using steel is the production of injection molding screws. The main concept for increasing the macro-hardness is the high content of MC and M_7_C_3_ carbides, which we can observe in the microstructure in Figure 1. All samples were etched with a special etchant designed for steels that are highly alloyed with Cr and Ni. The composition of the etchant consisted of 10 mL HNO_3_, 10 mL H_2_O_2_, 20 mL HCl, and 20 mL glycerin [47]. Thanks to the increased occurrence of carbide particles in the M398 steel, we can state that it would be possible to create screws enabling the processing of plastics with increased content of glass fibers [48].

The chemical composition of the experimental sample was determined and compared with the data from the steel manufacturer using spectral analysis on a SPECTROMAXx LMX10 device. The results of the spectral analysis, together with the chemical composition values provided by BÖHLER, can be seen in Table 1.

The material in its unheated state is suitable for machining; it has favorable cutting conditions thanks to the content of elements such as Mo and Mn.

With the chemical composition and method of steel production using powder metallurgy, the primary goal of which is to ensure a fine-grained structure, we expect that the steel will already have high strength and low susceptibility to plastic deformation in its basic state [49,50]. In order to determine the ultimate tensile strength Rm, the measurement was carried out on the INSTRON 5500R tensile test machine. For the measurement, samples were made according to Figure 2b. After the measurement, tensile curves were generated, which, as expected, do not have a significant yield strength. The highest average measured value of the tensile strength reached 1078.5 MPa (Figure 2a); this value was used as input data for later simulated calculations in the JJMatPro^®^ API v7.0 simulation software.

#### 2.1.1. Manufacturing Process

The BÖHLER company uses the powder metallurgy process, which makes it possible to create steel with a very fine distribution of carbides and alloying elements. The production process consists of melting the base material in an induction furnace. To ensure high homogeneity, the metal is mixed with the help of electromagnetic mixing. Molten metal is sprayed into a chamber with a protective nitrogen atmosphere, which enables the creation of a fine metal powder [51,52,53,54]. Rapid cooling enables the creation of a highly homogeneous powder with a fine dispersion of several types of carbides. Subsequently, the powder is placed in thin-walled cylindrical containers, and with the help of welding, the containers are closed and placed in an isostatic press. In the press itself, the connection of the individual powder particles takes place under the action of a pressure of 1000 bars and a temperature of 1150 °C. The combination of high pressure and temperature will allow the particles to fully bond without additional porosity. After cooling, there is a noticeable reduction in the height and diameter of the containers by roughly 10%. The result is a special tool steel whose microstructure has very fine carbides evenly distributed throughout the entire volume of the material.

#### 2.1.2. Phase Fraction

Based on the chemical composition, a phase diagram simulating the phase ratio for a given temperature was created with the help of JMatPro^®^ API v7.0 simulation software (Figure 3). The resulting phase diagram provides approximate information about the critical temperatures at which individual phase transformations occur. Determining these temperatures is a useful tool for further analysis of the examined steel from the point of view of dilatometric measurements. However, the diagram created using the JMatPro simulation program is only an approximate tool for determining phase fractions, as the principle of its creation is based on simulating the entire spectrum of temperatures from 0 °C to 1600 °C; this range represents a continuous temperature change from solidus to liquidus. In the case of M398 steel, due to the production process, a more complex temperature cycle occurs, which was described in more detail in the previous chapter.

The entire temperature regime of sample processing on the dilatometric device is shown in Figure 4. The temperature regime consists of heating to the austenitizing temperature of 1150°C at a rate of 1 °C/s and subsequent cooling of the samples at a selected rate in the temperature spectrum from 100 °C/s to 0.01 °C/s.

## 3. Results and Discussions

### 3.1. Determination of Critical Transformation Temperatures during Heating Ac_1_ and Ac_3_

The result of heating the studied sample of M398 steel according to the specified parameters is the dilation curve shown in blue in Figure 5. Since the dilatation curve does not have significant step changes, it is necessary to create a derivative of the dilatation curve according to time in order to accurately determine the austenitization temperatures [55,56]. On the resulting derivation curve shown in black, we can determine the beginning and end of the transformation of the BCC lattice to FCC austenite, i.e., temperatures Ac_1_ and Ac_3_. On the derivative curve, we can also observe another step change at a temperature of ~700 °C. The given step change represents the dissolution process of M_7_C_3_ carbides. At a temperature of ~700 °C, there is also a change in the magnetic properties of the examined sample from ferro-magnetic to paramagnetic; this phenomenon is observable as a sudden change in the power of the coil. This power change directly points to the pass through the Curie temperature [57,58].

For this reason, the curve representing the power of the heating coil is shown in green. A significant increase in the power of the heating coil occurred at a temperature of 680 °C; this temperature clearly points to the Curie temperature. The deviation appearing on the dilatation curve and, more significantly, on the derivative curve at a temperature of 710 °C represents diffusion structural changes of the primary carbides M_7_C_3_.

The temperature of Ac_1_ and Ac_3_ was determined using dilatometric measurement under heating conditions at a rate of 1 °C/s. However, the height of temperatures Ac_1_ and Ac_3_ changes depending on the heating rate, as shown in Figure 6, in which we can see the increasing trend of temperatures Ac_1_ and Ac_3_ with increasing heating rate [59,60,61,62].

### 3.2. Curie Temperature (TC) Measurement 

The Curie temperature (TC) is a basic quantity in the study of magnetic materials. It corresponds to the temperature at which a magnetically ordered material becomes magnetically disordered, i.e., it becomes paramagnetic. The Curie temperature also signifies the strength of the exchange interaction between the magnetic atoms. Figure 7 shows the variation of the initial permeability (*µ*’) with temperature for the M398. It is seen that the initial permeability increases with temperature up to the TC. This result could be explained according to the Globus et al. relation [63], which is given by:(1)$$\mu^{\prime }=\frac{M_{s}^{2}D}{\sqrt{\kappa_{1}}}$$
where *D* is the average grain size, *M_s_* is the saturation magnetization, and *κ*_1_ is the anisotropy constant. The variation of *µ*′ with temperature can be expressed as follows: The anisotropy constant (κ_1_) and saturation magnetization (*M_s_*) usually decrease with an increase in temperature. It is known that the anisotropy constant usually decreases much faster with temperature than with saturation magnetization [64], which leads to the increase in *µ*′. The maximum value of *µ*′, just below the TC, is a manifestation of the Hopkinson peak that is attributed to the minimization of anisotropy energy with temperature. Beyond this peak value *µ*′, the initial permeability sharply falls to a very low value indicating the ferro-paramagnetic phase transition. TC has been taken as the temperature at which a sharp fall of permeability is observed, i.e., where dµ′/dT attains its maximum value [65].

### 3.3. Dilatation Curves Analysis

Out of all eight cooling cycles, three (100, 1, and 0.001 °C/s) were selected, during which the largest quantitative changes occurred and on which an in-depth analysis of the change in dilation behavior and phase transformations was performed. At a cooling rate of 100 °C/s (Figure 8a) is the dilatation curve shown in blue, its derivative in orange for a more accurate determination of the Ms value, which was reached at a temperature of 246 °C. At this rate, we did not expect the formation of a microstructure other than a purely martensitic one. These experimental measurements were compared with JMatPro^®^ API v7.0 simulation software to compare physically measured values. A sudden change in the linearity of the dilatation curve, indicating the breakdown of the K12 austenite lattice into martensite, occurred according to the software at a temperature of 320 °C (Figure 8b). At the same time, it was possible to simulate the proportion of individual phases after simulated heat treatment. The results are shown in Figure 8c, where we can clearly see that at a temperature of 1280 °C, there should be a transition from liquid to austenite γ and, subsequently, the breakdown of γ to a martensitic microstructure. The simulation includes only the initial temperature Ms, as for this type of steel (which contains 2.7% C), the temperature Mf is below 0 °C.

Figure 9a shows the experimentally measured dilatation curve with a cooling rate of 1 °C/s compared to the simulated dilation curve (Figure 9b). It is clearly seen that two step changes occur immediately after each other. We assume that the first step change at a temperature of 450 °C represents a bainitic transformation, and then the second step change represents a transformation to martensite at a temperature of 211 °C. This temperature representing Ms is slightly lower compared to cooling at a rate of 100 °C/s; the phenomenon accompanying the drop in Ms temperature with decreasing temperature is described in more detail in the literature [66,67]. The simulated proportion of phase fractions provides a more detailed overview of the individual phases’ ratio in the resulting microstructure (Figure 9c). Despite the slower cooling rate, only a small amount of bainite occurs in the microstructure at a cooling rate of 1 °C/s; with a further decrease in the rate, we assume the formation of a pearlitic phase, which leads us to the conclusion that M398 steel is not suitable for bainitic anisothermal hardening. In order to achieve an increase in the proportion of the bainitic structure, it would probably be necessary to include isometric hardening to suppress the formation of the martensitic phase [68,69].

As we assumed, and we can also see in Figure 10a, in the temperature range of 200–500 °C, there are no step changes, as at 750 °C, a diffuse transformation of γ to pearlite occurred. This transformation can also be seen on the simulated dilatation curve (Figure 10b), where a step change occurred at 780 °C. The last of this series of images is represented by Figure 10c, which depicts the simulated proportion of phase fractions; it is clearly seen that the resulting microstructure will be formed by pearlite.

### 3.4. CCT Diagram

Figure 11 presents a comparison of two CCT diagrams. The diagram shown with solid lines represents the actual measured CCT diagram of M398 steel made based on the evaluation of cooling curves from dilatometric measurements. The dashed line shows the diagram, which is made based on simulated cooling curves from the JMatPro^®^ API v7.0 simulation software. Both diagrams are inserted into one figure for a simplified comparison of measured and simulated results. Cooling rates were determined during physical measurement and simulation in the range of rates from 100 °C/s to the slowest 0.01 °C/s. A total of eight dilation curves were evaluated for each pair of diagrams. Both produced CCT diagrams clearly demonstrate the presence of martensitic, bainitic, and pearlitic phases, while the simulated proportion of individual phases for selected cooling rates can be seen in Figure 12. Temperatures Ac_1_ and Ac_3_ were determined as an average value from all dilatometric measurements. The temperatures Ac_1_ and Ac_3,_ marked with dashed lines, were provided automatically by the calculation software when the cooling simulation was performed. These values of the austenitization temperature were unchanged from the cooling rate. In the case of actual experiments, the temperatures of Ac_1_ and Ac_3_ vary slightly depending on the heating rate. This issue is dealt with in more detail [70]. From the diagram, we can determine the measured temperatures of Ac_1_ = 960°C and Ac_3_ = 1070°C. The simulated values of austenitization temperatures are Ac_1_ = 962 °C and Ac_3_ = 1016 °C.

Based on our measurements and simulation, a fully martensitic microstructure can be achieved at cooling rates approaching 100 °C/s, since at a cooling rate of 10 °C/s we noted a step change in the expansion curves, indicating the formation of the bainitic phase, but the simulation with the help of software JMatPro does not predict an amount of bainite greater than 0.02%. The Ms curve, which was constructed based on dilatometric measurements, has a decreasing tendency, while at a cooling rate of 100 °C/s, the temperature is Ms = 243 °C, and subsequently, with a gradually decreasing cooling rate, the Ms temperature drops to a value of Ms 195 °C at a cooling rate of 0.5 °C/s. At lower rates of cooling, we assume a structure formed by pearlite and bainite. For this reason, the Ms curve at the given speed of 0.5 °C/s is already finished, and at the next lower speed of 0.1 °C/s, the martensitic change on the dilation curve was no longer detected. The calculation software Jmat determined the value of Ms to be a constant value of 335 °C in the entire range of quenching rates. The simulated temperature Ms, and Bs and Ps, are generally higher than the actual measured values by ~80 °C upwards. This phenomenon is probably caused based on the M398 steel production technology. The JMatPro^®^ API v7.0 simulation software for determining and correctly correcting the individual temperatures depends on a predefined grain size, but M398 steel has a very fine macrostructure that is unattainable using conventional foundry methods. For that reason, it was not possible for the software to accurately define the correct grain size, which was reflected in the upward shift of the boundary temperatures in the entire spectrum. However, in view of the initial cooling rates, these rates, regions B and P, are almost identical.

At cooling rates of 10, 5, 1, and 0.5, we noted the presence of both phase transformations, namely martensite and bainite, on the expansion curves made with the help of a dilatometer. At the same time, Bs temperatures increased with decreasing cooling rates. Overall, Bs temperatures ranged from 215 °C at a cooling rate of 10 °C/s to 357 °C at a cooling rate of 0.5 °C. The simulation assumes a bainitic transformation even at a cooling rate of 0.1 °C/s. However, in terms of phase abundance at this cooling rate, we assume a significantly higher amount of pearlitic phase based on the simulated phase fraction results. We assume the presence of a pearlitic phase based on simulation and measured results from a cooling rate of 0.5 °C/s. At the same time, the amount of phases increases by leaps and bounds with the decreasing cooling rate. At cooling rates of 0.05 and 0.01 °C/s, we assume a fully pearlitic–carbide microstructure. Lower cooling rates were not tested because they are not important from a technological and practical point of view. The temperature Ps ranges from 620 to 720 °C based on the measured expansion curves. In the case of simulation, the values are approximately 50–80 °C higher.

### 3.5. Hardness in CCT Diagram

Comparing the hardness of the measured experimental samples with the hardness from the JMatPro^®^ API v7.0 simulation software shows significant differences in values at the fastest cooling rates (Figure 13). The highest hardness value of 846 HV10 was achieved at a cooling rate of 100 °C/s. The JMatPro^®^ API v7.0 simulation software indicates that the hardness achieved at the given rate is 637 HV, a difference of more than 200 HV. This hardness value of 637 HV, according to JMatPro^®^ API v7.0 simulation software, has the same value up to a cooling rate of 1 °C/s. However, this rate is still lower than the actual measured samples for a given rate of 1 °C/s. Increasing the proportion of the bainitic structure leads to a gradual decrease in hardness in the speed range of 100 °C/s–0.5 °C/s. A significant decrease in hardness occurred at a cooling rate of 0.1 °C/s, which is associated with the appearance of a large proportion of the pearlitic structure and a small proportion of the bainitic structure. The measured hardness had a value of 337 HV10, while the hardness from the JMatPro^®^ API v7.0 simulation software maintained its value at the level of 587 HV as with the previous cooling rate of 0.5 °Cs. From a cooling rate of 0.1 °C/s, we observe that the JMatPro^®^ API v7.0 simulation software shows higher hardness values compared to the actually measured samples. The lowest hardness of 284 HV10 was measured on the last sample at a cooling rate of 0.01 °C/s, where the volume of the material was formed only by a purely pearlitic structure. Overall comparison of the hardness analysis shows that with the change in the cooling rate, there is also a change in the hardness of the measured samples [71,72,73]. This decrease is associated with microstructural changes that occurred after individual cooling rates, as well as changes from a homogeneous single-phase microstructure to a multiphase heterogeneous microstructure formed by bainite and pearlite [74,75].

### 3.6. Microstructure in CCT Diagram

The microstructure analysis of the selected three cooling rates is shown in Figure 14. The samples were prepared by standard according to ASTM E3; the preparation process included wet sanding with silica papers with a grain size of 240, 500, 800, and 1200 μm. Subsequently, the samples were polished using diamond abrasion with a grain size of 9, 6, 3, and 0.5 μm under a load of 25 N, 10 min, and 150 rpm for each selected abrasive. The microstructure after the highest cooling rate of 100 °C/s consisted exclusively of martensitic matrix and large M_7_C_3_ carbides, and smaller MC carbides (Figure 14a). Primary austenitic grain boundaries (PAGB) are also faintly recognizable on the given microstructure. The microstructure of the sample cooled at a rate of 1 °C/s (Figure 14b) also consisted of a martensitic matrix, which, according to the dilatometric curve, also contained a smaller content of the bainitic phase. However, this phase is very difficult to detect by microscopic analysis of the given powder steel. This structure also contains carbide particles M_7_C_3_ and MC, mainly based on Cr and V. The last examined sample, which was cooled at a rate of 0.01 °C/s (Figure 14c), shows an exclusively pearlitic structure, which is formed by slightly globularized cementite particles. Carbide particles occur mainly at the boundaries of PAGB, which are clearly recognizable.

The determination of the chemical composition was carried out with an EDX detector, which captures the characteristic X-rays emitted from the analyzed area as a result of the effect of the electron beam of the thermoemission scanning electron microscope on the surface of the material being examined [76]. The energy-dispersive spectroscopy (EDS) method evaluates the obtained X-ray spectrum and processes it into a signal, subsequently monitored as an energy spectrum [77]. Due to the effect of applying three cooling rates of the M398 steel, transformations of the distribution, size, and partly also the shape of the complex carbide phase occur. Changes in the chemical compositions obtained from the analyzed areas are also related to this process. There is a change in the ratio of the representation of the steel matrix and complex carbides. This change is also reflected by the increase in the content of the chemical element Fe towards a lower cooling rate, as such conditions offer more space for the diffusion of carbide-forming elements (Cr, V, W, Mo) and for the formation of a smaller number of complex carbides of larger dimensions [78]. The distribution of the carbide phase is less uniform when the cooling rate is reduced, which results in a greater representation of the metal matrix in the analyzed area (Figure 15). The working conditions of the EDS method are not suitable for the exact determination of the C content in steel [79]; therefore, it is appropriate to interpret the results of the chemical analysis of carbon as proportional (relative change in the carbon content at individual steel cooling rates).

In addition to C, important carbide-forming elements such as Cr and V, whose significant agglomeration is concentrated mainly in the occurrence of complex carbides, participate in the transformations of the carbide phase (Figure 16). Significant diffusion of these elements in the metal matrix in connection with lower cooling rates of steel can be directly related to changes in the distribution and size of the carbide phase (Figure 16a–c). The chemical element W does not show the ability to agglomerate as significantly under the selected conditions of steel cooling as the elements mentioned above. The chemical element Mo also records the ability to form microregions with its more intense occurrence but at higher cooling rates (Figure 16d). However, its agglomeration is represented to a much smaller extent than with the elements C, Cr, and V. At lower cooling rates of the steel, the ability of the agglomeration of the element Mo is significantly suppressed, and instead, it appears that under such conditions the distribution of molybdenum is more or less uniform.

Nanoindentation analysis was performed on prepared metallographic samples at different cooling rates with the application of internal Berkovich geometry at a standard room temperature of 23 °C, where the output measured values were the values of the reduced Young’s modulus of elasticity Er [GPa] and nanohardness H [GPa]. The experimental method for steel type M398 was carried out using the Hysitron Triboindenter TI 950 working device with the evaluation software Triboscan. Individual areas of research were determined based on an optical microscope.

As part of the testing, three measurements were carried out at specific speeds of 100, 1, and 0.01 °C/s with different distributions of indents at the specified location of the microstructure. As part of the testing, a standard trapezoid was used with a maximum load of 8000 µN for a period of t = 2 s; tests were carried out in accordance with the ISO 14,577 standard. The location of specific indents for the selected speeds is clearly shown on the SPM scan with dimensions of 50 × 50 µm (Figure 17). An overview of the values of nanohardness H and reduced Young’s modulus of elasticity Er obtained by nanoindentation analysis are compared in Figure 18.

The measurement was performed on in situ SPM (scan probe microscopy) scans of selected locations. In all three cases, the microstructure of the SPM scan consists of hard particles (carbides) and a matrix. Since each sample underwent different cooling conditions, the microstructure of the matrix differs from one another. The mentioned differences were also confirmed by nanoindentation measurements. The highest nanohardness value was achieved by the matrix in the case of cooling at 100 °C/s, where there is a high proportion of the pure martensitic phase, and the lowest in the case of a cooling rate of 0.01 °C/s, where a heterogeneous mixture of pearlite prevails (Figure 18a). The values of the reduced modulus of elasticity also correlate with the given assumption. The martensite-dominated matrix has a higher modulus of elasticity than the pearlitic–bainitic structure (Figure 18b). The nanohardness of the carbide particles showed a decrease in both the nanohardness value and the modulus of elasticity in the case of cooling at 1 °C/s. The reason may be the formation of secondary carbides of a different type from primary carbides during bainitic transformation. At the fastest cooling of 100 °C/s, diffusion processes do not take place; therefore, only primary carbides formed during primary crystallization remain in the structure. Conversely, during very slow cooling, some carbides dissolve, and only primary carbides remain in the structure. At the same time, with the slowest cooling, the proportion of the number of carbides and the area of the matrix is the smallest, and vice versa.

## 4. Conclusions

The article deals with the dilatometric behavior of powder tool steel M398 at eight cooling rates from 100 °C/s to 0.01 °C/s. The analysis of dilatation curves led to the construction of the final CCT diagram, which was subsequently compared with the CCT diagram from the JMatPro^®^ API v7.0 simulation software. The next part of this paper examined the resulting microstructural analysis using SEM and EDS map analysis as well as tracking the change of nanoindentation properties using AFM. From the obtained results, we can present the following conclusions:Setting the austenitizing temperature to 1120 °C led to complete austenitization of the entire volume of the sample material, while the temperatures Ac_1_ and Ac_3_ were 960 °C and 1070 °C, respectively;Heating the samples at a rate of 1 °C/s provides ideal properties due to the optimal setting of the combination of output mechanical and economic properties. Higher heating rates lead to an increase in the temperature Ac_1_ and Ac_3,_ respectively (increasing PAGS), and conversely, a decrease in temperature leads to an extension of the sample heating time and, thus, to an economic increase in operating costs;The actual measured CCT constructed for M398 steel consisted of three different microstructural regions, namely martensitic, bainitic, and pearlitic. The resulting CCT diagram can be used to design various heat treatments. The critical cooling rate must be set to more than 10 °C/s in order to obtain a purely martensitic structure. To reduce the amount of residual austenite, it is necessary that the temperature Mf moves up to negative temperatures due to the high proportion of C in the M398 material;The simulation software JMatPro^®^ API v7.0 is a good tool for the approximate determination of the CCT diagram, but it does not present as qualified results as the actual dilatometric measurements. The temperature difference Ms of almost 100 °C is an important factor that confirms this assumption;The measured hardness has a decreasing character depending on the decrease in the cooling rate. The highest hardness achieved was 846 HV10 at the highest cooling rate of 100 °C/s, while the lowest hardness was 284 HV10 at the lowest cooling rate of 0.01 °C/s;Experimental steel is not suitable for bainitic quenching during anisotemic cooling due to the occurrence of only a small proportion of this structure. An increase in the volume of the bainitic phase would be possible only during isothermal hardening;The distribution of the carbide phase is less uniform when the cooling rate is reduced, which results in a greater representation of the metal matrix in the analyzed area. The chemical element W does not show the ability of such significant agglomeration under the selected cooling conditions as the above-mentioned elements. Molybdenum records the ability to form microregions, but only at higher cooling rates;The highest value of nanohardness was achieved by the matrix in the case of cooling at 100 °C/s, where there is a high proportion of the pure martensitic phase, and the lowest in the case of a cooling rate of 0.01 °C/s, where a heterogeneous mixture of pearlite (ferrite + cementite) prevails. The values of the reduced modulus of elasticity also correlate with the given nanohardness results.

From the overall point of view, we can conclude that the constructed CCT diagram plays an important role in setting the quenching process of experimental powder steel M398. An important factor is also the subsequent determination of the secondary heat treatment in the form of the tempering regime (tempering temperatures, endurance at the tempering temperature, use of deep freezing, and the number of tempering cycles), by which we can achieve a change in mechanical properties in the form of secondary hardening. This process will be explored in more detail in the next part of the research.

## Figures and Tables

**Figure 1 materials-16-04473-f001:**
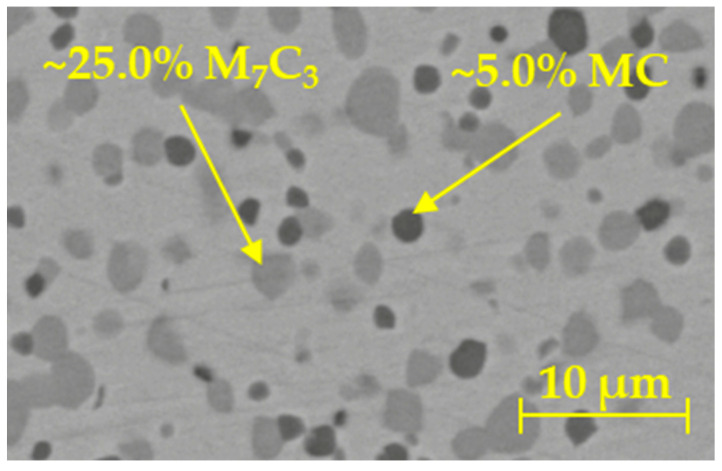
Microstructure of base material M398.

**Figure 2 materials-16-04473-f002:**
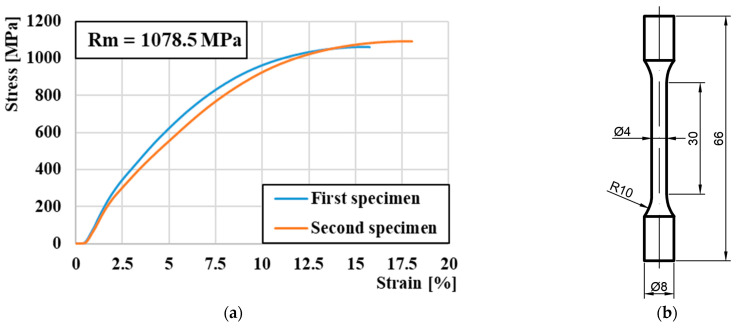
(**a**) Stress-strain diagram of M398 steel. (**b**) The shape of the tensile test specimen (mm) by ASTM E8/E8M-13.

**Figure 3 materials-16-04473-f003:**
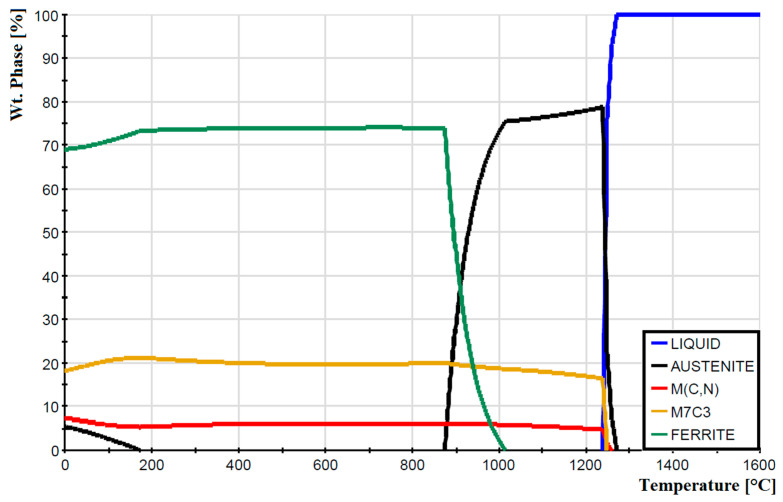
Phase fraction JMatPro of powdered tool steel M398.

**Figure 4 materials-16-04473-f004:**
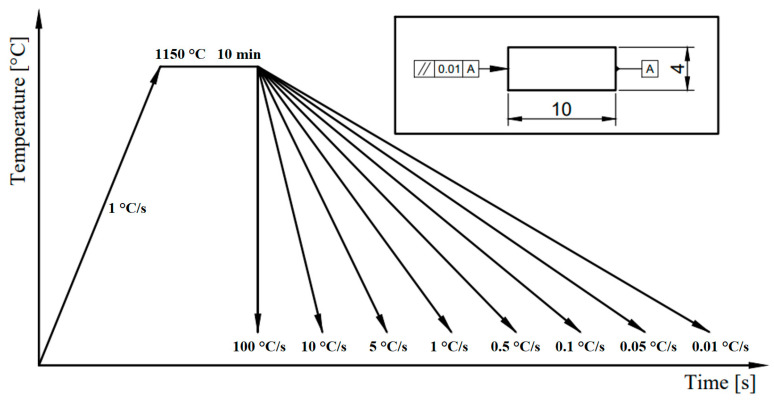
Thermal modes used for dilatometric analysis.

**Figure 5 materials-16-04473-f005:**
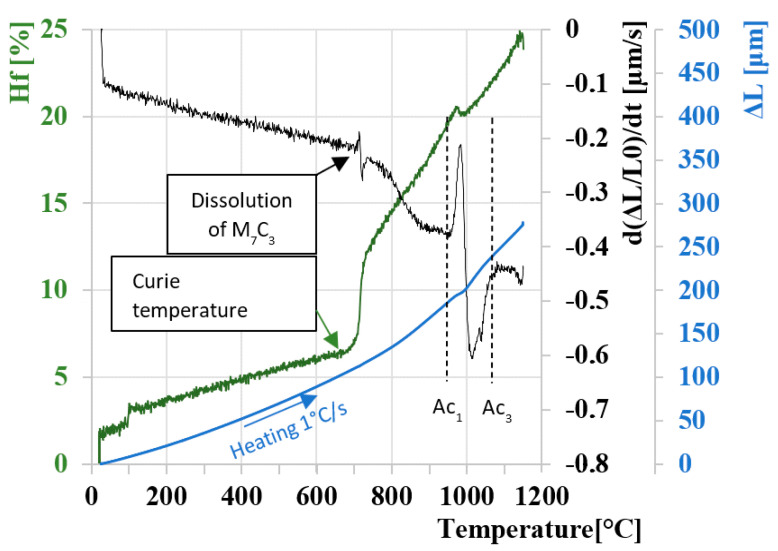
Analysis of dilatation curve during heating 1 °C/s.

**Figure 6 materials-16-04473-f006:**
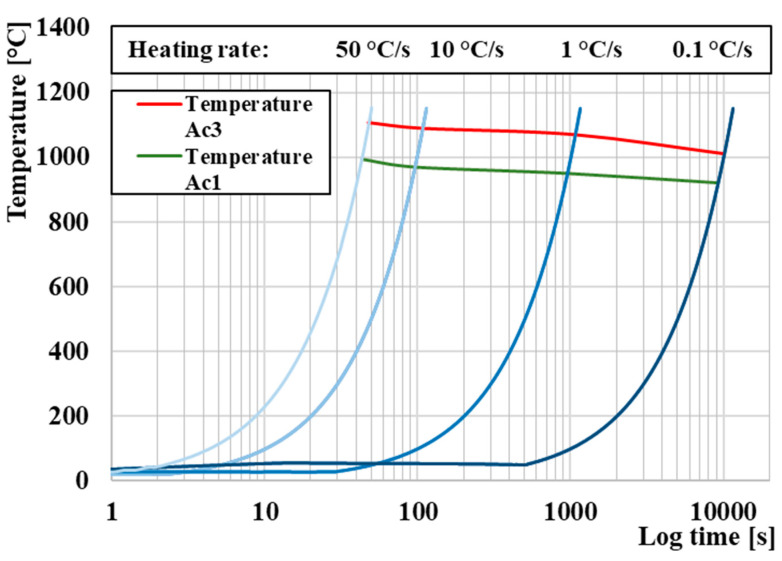
Effect of heating rate on temperatures Ac_1_ and Ac_3_.

**Figure 7 materials-16-04473-f007:**
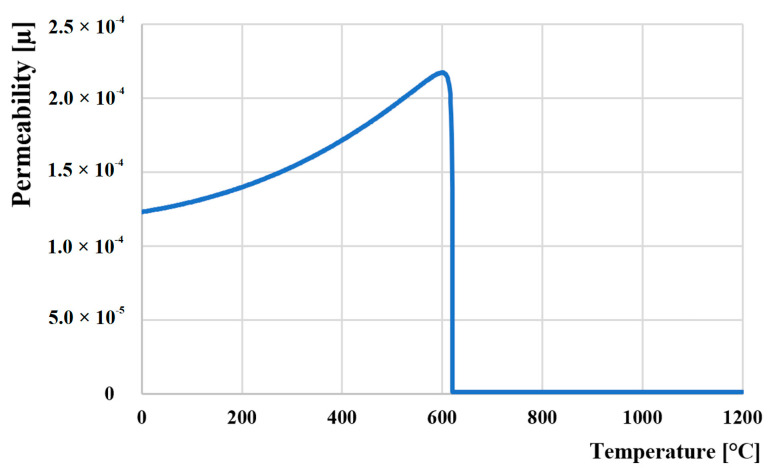
Change in permeability during heating of material M398 (heating 1 °C/s).

**Figure 8 materials-16-04473-f008:**
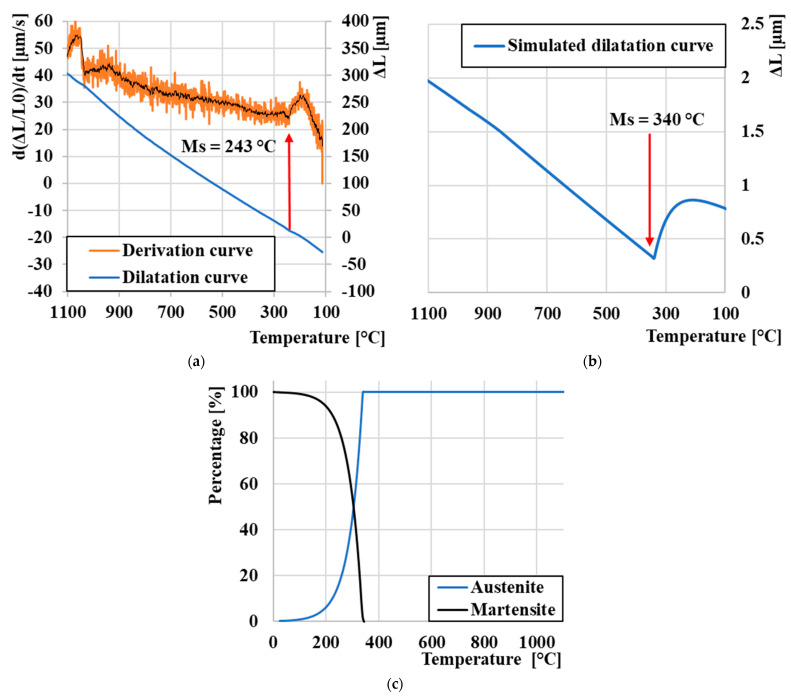
Analysis of the expansion curve: (**a**) experimental measurement cooling 100 °C/s, (**b**) cooling 100 °C/s JmatPro, and (**c**) proportion of phase composition JMatPro.

**Figure 9 materials-16-04473-f009:**
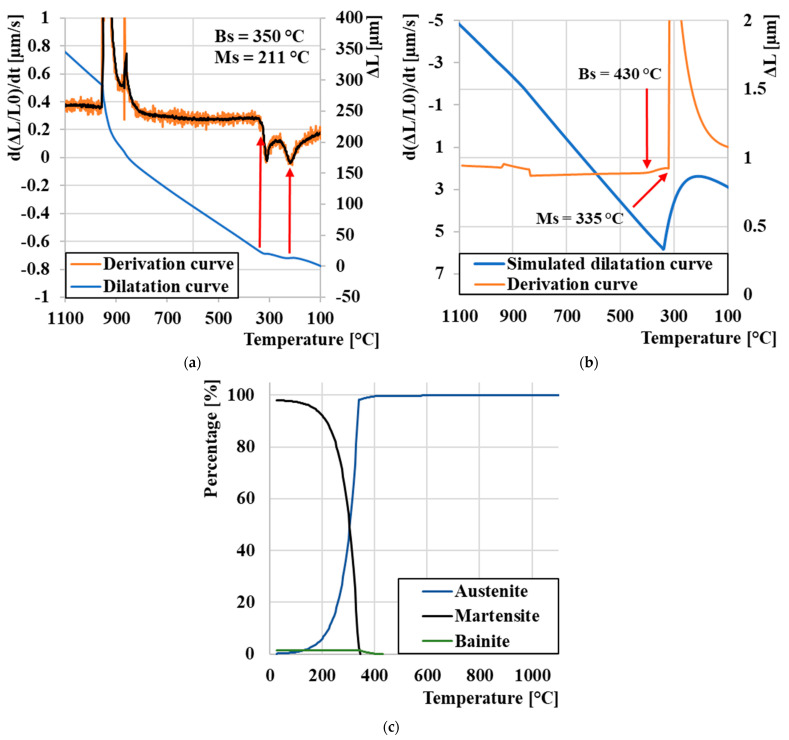
Analysis of the expansion curve: (**a**) experimental measurement cooling 1 °C/s, (**b**) cooling 1 °C/s JMatPro, and (**c**) proportion of phase composition JMatPro.

**Figure 10 materials-16-04473-f010:**
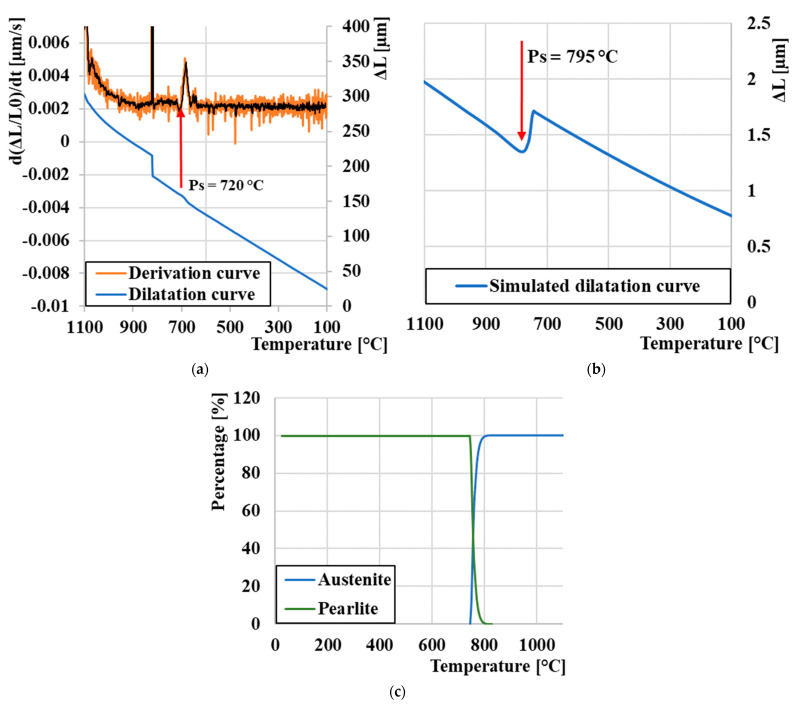
Analysis of the expansion curve: (**a**) experimental measurement cooling 0.01 °C/s, (**b**) cooling 0.01 °C/s JmatPro, and (**c**) proportion of phase composition JMatPro.

**Figure 11 materials-16-04473-f011:**
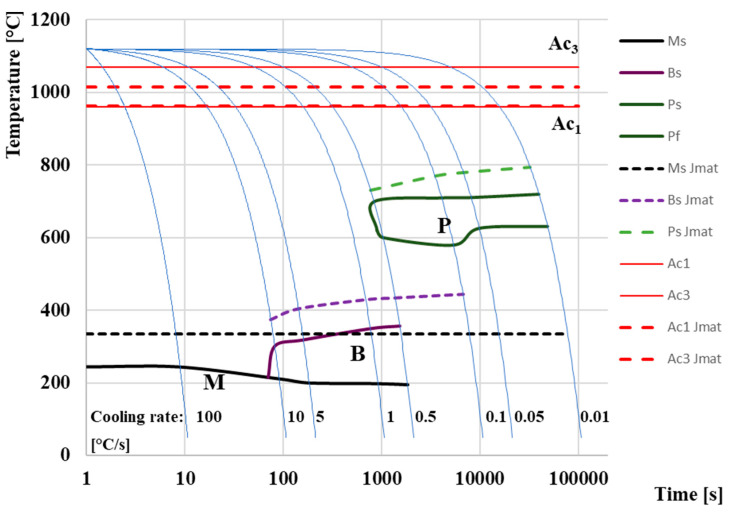
Comparison of CCT diagrams of powder tool steel M398.

**Figure 12 materials-16-04473-f012:**
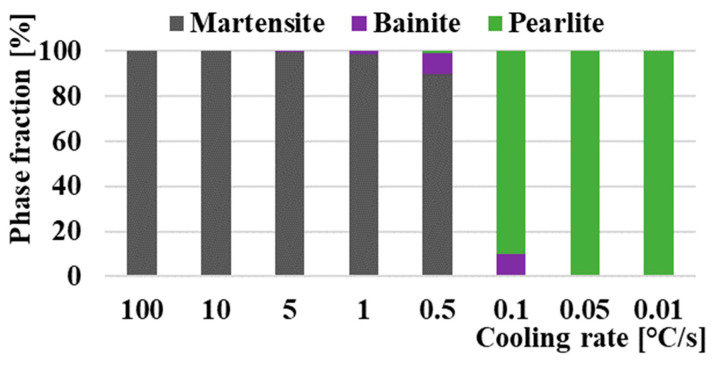
Simulated proportion of individual phases from JMatPro.

**Figure 13 materials-16-04473-f013:**
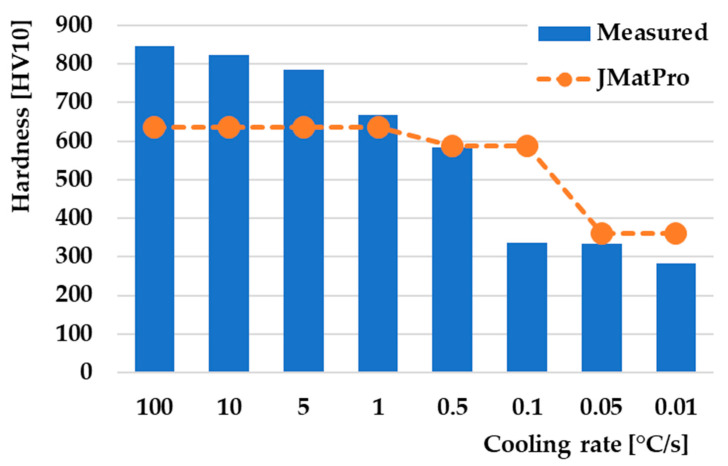
Comparison of hardness dependence on cooling rate.

**Figure 14 materials-16-04473-f014:**
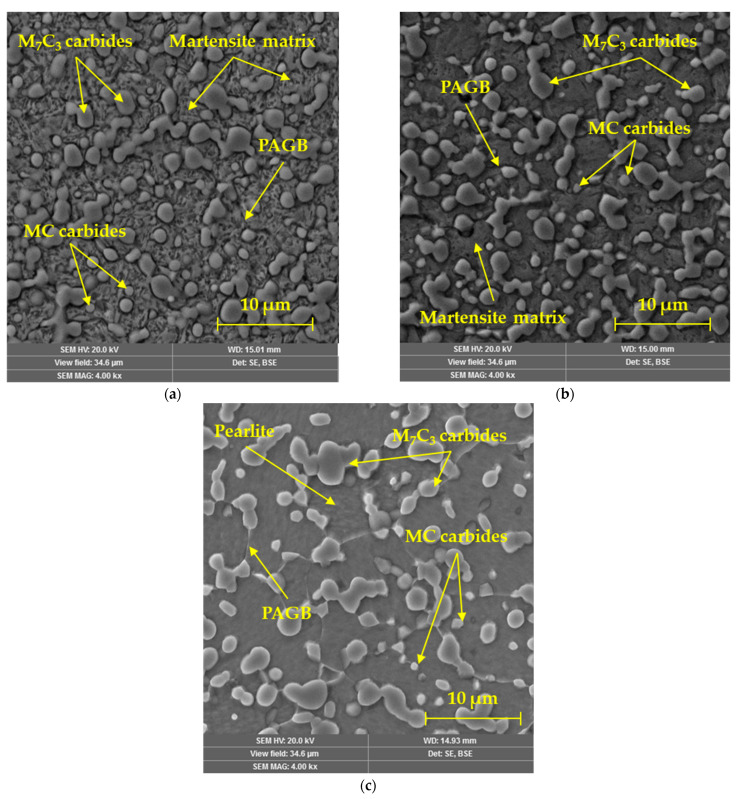
Microstructure of specimens obtained by SEM: (**a**) 100 °C/s, (**b**) 1 °C/s, and (**c**) 0.01 °C/s.

**Figure 15 materials-16-04473-f015:**
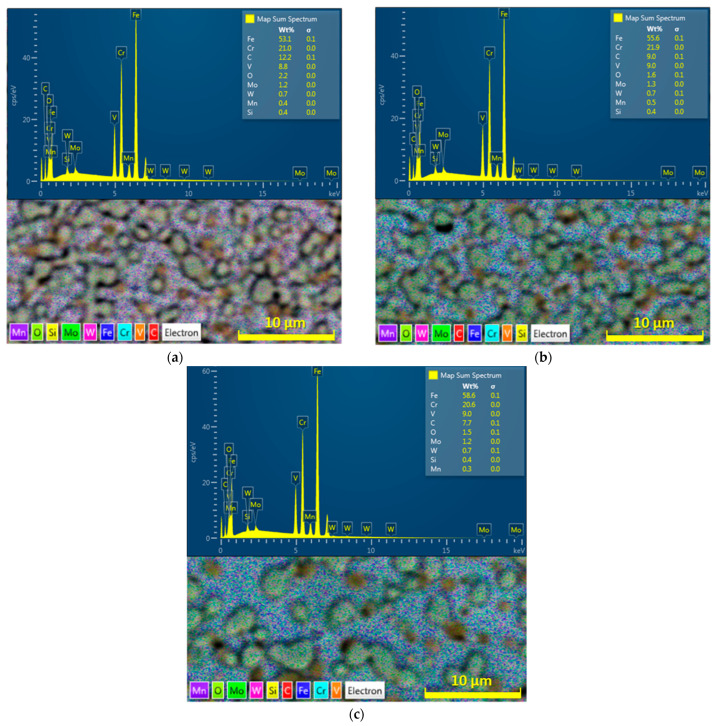
EDS layer image and map spectrum for cooling rate: (**a**) 100 °C/s, (**b**) 1 °C/s, (**c**) 0.01 °C/s.

**Figure 16 materials-16-04473-f016:**
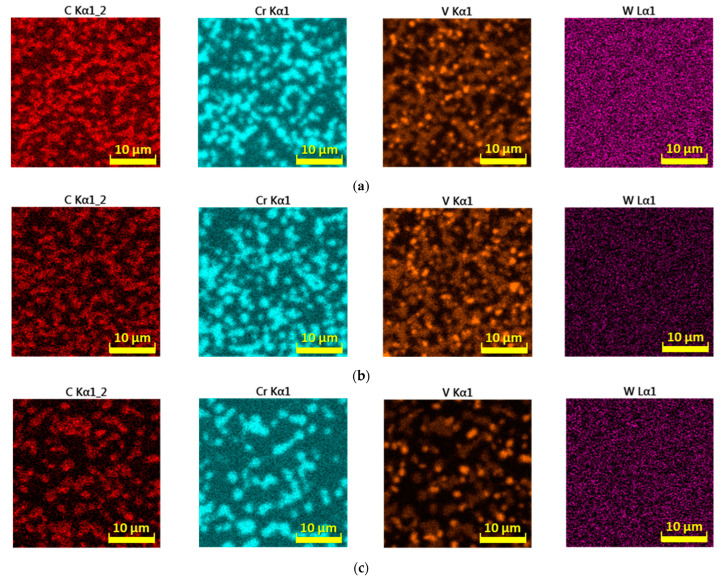
EDS maps of selected elements: (**a**) 100 °C/s, (**b**) 1 °C/s, (**c**) 0.01 °C/s, (**d**) EDS maps of molybdenum.

**Figure 17 materials-16-04473-f017:**
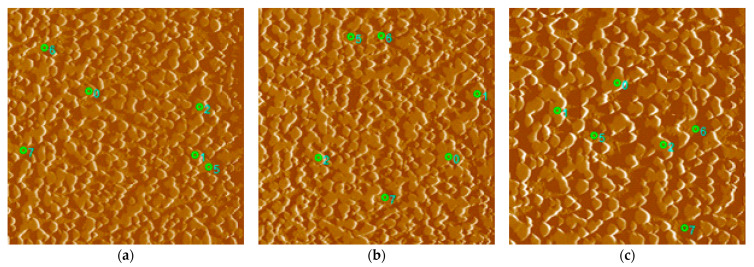
SPM scan: (**a**) 100 °C/s, (**b**) 1 °C/s, (**c**) 0.01 °C/s. Indents no. 0; 1; 2 = carbides; 5; 6; 7 = matrix.

**Figure 18 materials-16-04473-f018:**
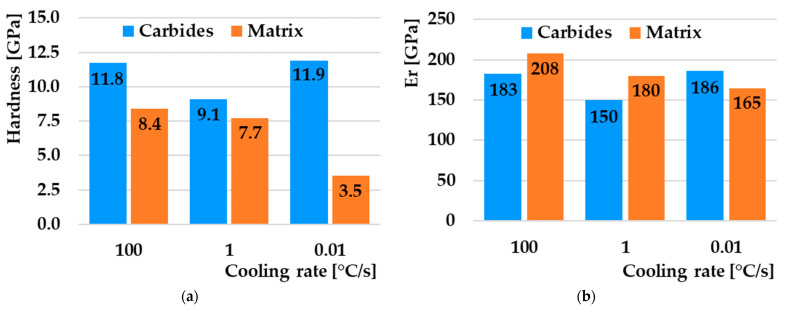
(**a**) Comparison of nanohardness. (**b**) Comparison of reduced modulus of elasticity.

**Table 1 materials-16-04473-t001:** Chemical composition of the investigated M398 steel (wt.%).

Element	C	Mn	Si	Cr	Mo	V	W
**Böhler**	2.70	0.50	0.50	20.00	1.00	7.20	0.70
**Spectral analysis**	2.72	0.50	0.51	20.07	1.00	7.22	0.70

## Data Availability

Data are available upon request to the corresponding author.

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
