# Peer review of "Microstructural Changes and Determination of a Continuous Cooling Transformation (CCT) Diagram Using Dilatometric Analysis of M398 High-Alloy Tool Steel Produced by Microclean Powder Metallurgy"

_materials, 2023, doi:10.3390/ma16124473_

Round 1

Reviewer 1 Report

Major comments:

1. In Section 2.2.1, the authors state the manufacturing process of the tool steel, but I can’t find any relationship with the context. Can the authors explain why the production process of this tool steel is presented here?

Minor comments:

1. Line 93, “Fig. 1” should be replaced by “Figure 1”.

2. Figure 2 (a), the unit of Rm is MPa, not Mpa.

3. In Figure 4, the unit of the X-axis should be corrected.

4. Line 200, Curie temperature (TC) should be replaced by TC.

5. Line 385, should be Figure 17, not 27.

Reviewer 2 Report

Review of the manuscript "Microstructural changes & determination of CCT diagram using dilatometric analysis of M398 high-alloy tool steel produced by microclean powder metallurgy" submitted for publication on Materials.

The manuscript is about a dilatometric study of high-alloy martensitic tool steel. The measured dilatation curves are compared with microstructural analysis. A comparison of surface hardness as well nanoindentation tests have been performed. 

The manuscript appears interesting while novelty aspects and possible applications should be better highlighted in the introduction and conclusion.

It can be reconsidered for publication at least after the following major revisions:

1) References in the list are not up to date, some of them quite ancient;

2) Non all references are indicated in compliance with the journal requirements (add doi when available, strictly follow author's instructions);

3) Line 43 [5-11], 7 references one shot. Line 53 [22-27] 6 references one shot. Unacceptable. Please explicitate each of them and motivate the citation. Check again the introduction to avoid such multiple references.

4) Figure 2 a) is not called in the main text;

5) Not Fig. 27 but Fig. 17 (line 385);

6) Fig. 13 is not called in the main text;

7) Chapter "Discussion" (core of a research article and based on the results) is not present. Please add!

After that the manuscript can be reconsidered for publication.

Reviewer 3 Report

The study focuses on the dilatometric analysis of a high-alloy martensitic tool steel (BÖHLER), i.e., the M398 designation, fabricated through powder metallurgy. The main objective of the study is to create a Continuous Cooling Transformation (CCT) diagram for the investigated powder steel at different cooling rates (~ 100 to 0.01 °C/s). The authors has used JMatPro simulation software to compare with the experimental CCT diagram. The dilatation curves obtained from the experiments are examined alongside microstructural analysis conducted using a scanning electron microscope (SEM). The analysis reveals the presence of carbide particles (M7C3 and MC) based on chromium (Cr) and vanadium (V). The distribution of selected chemical elements is evaluated using Energy-Dispersive X-ray Spectroscopy (EDS). Additionally, the study compares the surface hardness of all samples in relation to the cooling rates. Subsequently, the individual phases and carbides' nanoindentation properties are investigated using an Atomic Force Microscope (AFM), where the nanohardness and reduced modulus of elasticity are evaluated for both the carbides and matrix.

The manuscript is well written and organized and shows interesting results and discussion points. However, the originality of work is not clear. Below is a summary list of suggestive revisions that might help improve the manuscript.

1.       The “Abstract” section needs to outline the work; and briefly present what is given in the manuscript by answering what, why, how the research work was carried out; and possibly with a very brief reference to the results. Abstract should be 200 words according to the journal guideline.

The abstract may need to include a few sentences to provide a clear problem statement followed by clear emphasis on the originality of the work. At its current condition, it reads just like a summary of results. It also needs a couple of sentences regarding the experimental methodology.

2.       The “Introduction” section needs to include a clear problem statement followed by clear emphasis on the originality of the work in the last paragraphs.

3.       The “Experiments and Methods” section needs some minor revisions listed below:

3.1.     The ISO/ASTM standard procedure citation for metallography and mechanical testing (tensile and nano hardness tests).

3.2.    The details of metallography (such as the griding SiC sandpaper grit sizes and the applied forces used) and the electron microscopy (such as the kV and working distance used for the SEM and EDS analysis).

4.       In the “Results and discussion” section, some figures need attention.

4.1.    In general, the number of figures (20) in the manuscript is a little too many for such short manuscript. I would suggest reducing it to about 12-15 by combining or removing the unnecessary ones.

4.2.    Figure 2(b) needs unit and ASTM/ISO standard procedure number.

4.3.    I would suggest rearranging the layout of Figure 3 more in a vertical way to have better visual of the schematic. At its current condition, they are not clearly conveying the message, different stages of PM fabrication process needs to be clearly shown.

4.4.    The quality of Figure 4 needs improvement.

4.5.    Figure 7 needs clear legend. The horizontal lines in the graphs are not explained in the graph nor in the caption.

4.6.    The total number of figures (21) seems to be too many for a journal article with only 22 pages length. I would suggest reducing the number of figures to between 10-12 figures.

Minor edits on English of the manuscript may needed.

Round 2

Reviewer 2 Report

Manuscript significantly improved, it can be accepted as it is.